# Foliar-feeding insects acquire microbiomes from the soil rather than the host plant

S. Emilia Hannula [1], Feng Zhu[1,2], Robin Heinen [1,3] & T. Martijn Bezemer [1,3]

Microbiomes of soils and plants are linked, but how this affects microbiomes of aboveground herbivorous insects is unknown. We first generated plant-conditioned soils in field plots, then reared leaf-feeding caterpillars on dandelion grown in these soils, and then assessed whether the microbiomes of the caterpillars were attributed to the conditioned soil microbiomes or the dandelion microbiome. Microbiomes of caterpillars kept on intact plants differed from those of caterpillars fed detached leaves collected from plants growing in the same soil. Microbiomes of caterpillars reared on detached leaves were relatively simple and resembled leaf microbiomes, while those of caterpillars from intact plants were more diverse and resembled soil microbiomes. Plant-mediated changes in soil microbiomes were not reflected in the phytobiome but were detected in caterpillar microbiomes, however, only when kept on intact plants. Our results imply that insect microbiomes depend on soil microbiomes, and that effects of plants on soil microbiomes can be transmitted to aboveground insects feeding later on other plants.

[1] Department of Terrestrial Ecology, The Netherlands Institute of Ecology NIOO-KNAW, Droevendaalsesteeg 10, 6708 PB Wageningen, The Netherlands. [2] Key Laboratory of Agricultural Water Resources, Hebei Key Laboratory of Soil Ecology, Center for Agricultural Resources Research, Institute of Genetic and Developmental Biology, The Chinese Academy of Sciences, 286 Huaizhong Road, 050021 Shijiazhuang, Hebei, China. [3] Institute of Biology, Section Plant Ecology and Phytochemistry, Leiden University, P.O. Box 9505, 2300 RA Leiden, The Netherlands. These authors contributed equally: S. Emilia Hannula, Feng Zhu, Robin Heinen, T. Martijn Bezemer. Correspondence and requests for materials should be addressed to T.M.B. (email: m.bezemer@nioo.knaw.nl)

Soil microbiomes harbor an extremely rich diversity of bacteria and fungi[1,2]. Plants also have microbiomes, and as they are rooted in the soil, a subset of the soil microbiome colonizes the roots[3,4]. Consequently, aboveground plant parts, such as stems and leaves, are inhabited by specific commensal, symbiotic or pathogenic bacteria and fungi that, at least partly, originate from the roots and soil[5,6]. Insects are also associated with a variety of microbes[7–10]. These microbes can act as pathogens causing diseases[11] or can be beneficial for defense, detoxification, or digestion of food[12–15]. Herbivorous insects ingest microorganisms that are present in the plant, and hence microorganisms that originate from the soil, via the plant[6], can be incorporated in the microbiome of the insect[16]. However, recent studies suggest that many of these microbes may not persist in the caterpillar gut[10]. Studies using animals other than insects have shown that an important part of the microbiome originates from non-dietary sources[17,18]. Moreover, several studies have shown that herbivorous insects can take up specific symbiont bacterial species from the environment, and also directly from the soil[19,20]. Whether herbivorous insect microbiomes as a whole are also influenced by the soil environment is unknown. An intriguing possibility is that changes in soil microbiomes can lead to changes in insect microbiomes and alter the performance of insects, mediated via the microbiome of the plant, or through direct soil-insect interactions.

Plants have aboveground and belowground parts and act as the primary providers of resources for most other aboveground and belowground dwelling organisms[21]. Moreover, an overwhelming amount of research over the past two decades has shown that plants are pivotal in mediating interactions between these aboveground and belowground organisms. For instance, root-associated organisms can influence foliar feeding insects on the same plant[22,23]. Plants also change the microbiome of the soil they grow in, and this depends on plant traits such as plant growth form (grasses and forbs) and growth rate[24,25]. Other plants that grow later in these conditioned soils, and the insects feeding on those plants, respond to the changes in soil microbiomes[25,26]. So far, most research has focused on the role of systemic changes in the chemical composition of aboveground and belowground plant parts[27]. The role of changes in plant and insect microbiomes in these aboveground-belowground interactions is poorly understood, and how this is influenced by plant-mediated changes in soil microbiomes is unknown.

We hypothesize that plant-mediated changes in soil microbiomes will affect microbiomes of caterpillars feeding on plants that grow later in these soils, through modifications of the microbiomes of their host plants. We expect that plant growth form and growth rate are important drivers of soil microbiomes and that these microbiomes will affect the root and subsequently the shoot microbiome of our test plant species (*Taraxacum officinale*; Asteraceae), eventually altering the caterpillar (*Mamestra brassicae*; Lepidoptera; Noctuidae) microbiome. We use two parallel assays (Supplementary Fig. 1) to disentangle the effects of the soil microbiome on the caterpillar microbiome mediated via the plant from the possible direct effects via the soil. Using these two parallel assays, we show that the microbiome of an aboveground insect herbivore is shaped not by the microbiome of its host plant, but directly by the microbiome of the soil its host plant grows in.

## Results

**Composition of soil, plant, and insect microbiomes.** Briefly, microbiomes in the soil, plant and insect compartments were characterized by Illumina MiSeq sequencing, using 16S rRNA and ITS2 regions (for bacteria and fungi respectively). Rhizosphere soil contained the highest diversity of both bacteria and fungi, and leaves were the least diverse compartments (Fig. 1a, b; Supplementary Fig. 2). We use two parallel assays (Supplementary Fig. 1) to disentangle if the microbial diversity in caterpillars is affected by plants or by soils. Caterpillars that were fed detached leaves had a significantly lower diversity of both bacteria and fungi in terms of absolute diversity and a lower number of fungal phyla and bacterial classes than caterpillars fed on intact plants (Fig. 1a, b; GLM: bacteria: $F = 7.56$, $P < 0.001$; fungi: $F = 8.11$, $P < 0.001$). Both for bacteria and fungi, the community structure found in caterpillars fed on intact plants and in caterpillars fed on detached leaves differed significantly (PERMANOVA: bacteria: $F = 30.05$, $R^2 = 0.19$, $P < 0.001$; fungi: $F = 43.11$, $R^2 = 0.25$, $P < 0.001$) and there was a little overlap between the two types of microbiomes (Fig. 1c, d). Remarkably, microbiomes of caterpillars kept on intact plants resembled those found in soils much more closely than microbiomes of leaves or caterpillars fed on detached leaves (Fig. 1c, d). There were no significant differences in microbiomes of leaves collected from plants that had caterpillars on them, and leaves from plants that were kept without caterpillars and that were used to collect leaves from for the detached plant assay (Fig. 1c, d).

Not only did the total microbial community composition differ between the caterpillars fed on intact plants and those fed on detached leaves, the composition in terms of phylum and class levels also differed. The bacterial phyla Actinobacteria and Chloroflexi, and the fungal classes Eurotiomycetes, Sordariomycetes, and Dothideomycetes, were more abundant in caterpillars fed on intact plants, while Betaproteobacteria and a group of unclassified fungal OTUs were more abundant in the caterpillars that fed on detached leaves (GLM: FDR adjusted $P < 0.05$ for all cases; Supplementary Fig. 3). The leaf microbiome consisted almost entirely of a group of unclassified fungal OTUs and members of the bacterial phylum Gammaproteobacteria (Supplementary Fig. 4 and 5), both groups were also found more commonly in microbiomes of caterpillars fed on detached leaves, thus explaining the observed clustering (Fig. 1c, d). Root microbiomes comprised a subset of the soil community, and especially Gammaproteobacteria, Firmicutes, Bacteroidetes, Sordariomycetes, Agaricomycetes and Glomeromycotina were enriched inside the roots (Fig. 1c, d; Supplementary Fig. 4, 5).

**Shared microbes between soils, leaves, and caterpillars.** Caterpillars fed on intact plants and detached leaves shared a common core microbiome which was also present in the leaves (20.3% of their microbiome) and in the roots (19.1%) (Fig. 2a–c), but also harbored unique microbes; 16.7% of the caterpillar microbiome was found only in caterpillars. This core microbiome of caterpillars consisted predominantly of Proteobacteria, Acidobacteria, Firmicutes, and unclassified fungi (Supplementary Figs 6, 7). Remarkably, for caterpillars fed on intact plants, a large proportion of the OTUs found in caterpillars, was also detected in the soil (75%; represented as numbers 1 and 4 in Fig. 2a). Microbiomes of caterpillars fed detached leaves had virtually no additional OTUs that were not also found in caterpillars kept on intact plants (Fig. 2c), but the microbiomes of the latter contained three times more OTUs. The main groups of shared OTUs between soils and caterpillars kept on intact plants were Actinobacteria (12.6% of OTUs), Eurotiomycetes (21.8%) and unclassified fungal OTUs (22.3%) (Supplementary Fig. 6). Furthermore, the fungal class Eurotiomycetes and bacterial phylum Actinobacteria were represented in a disproportionally high ratio in caterpillars that were kept on intact plants, compared to their abundance in soil (Supplementary Fig. 4, 5).

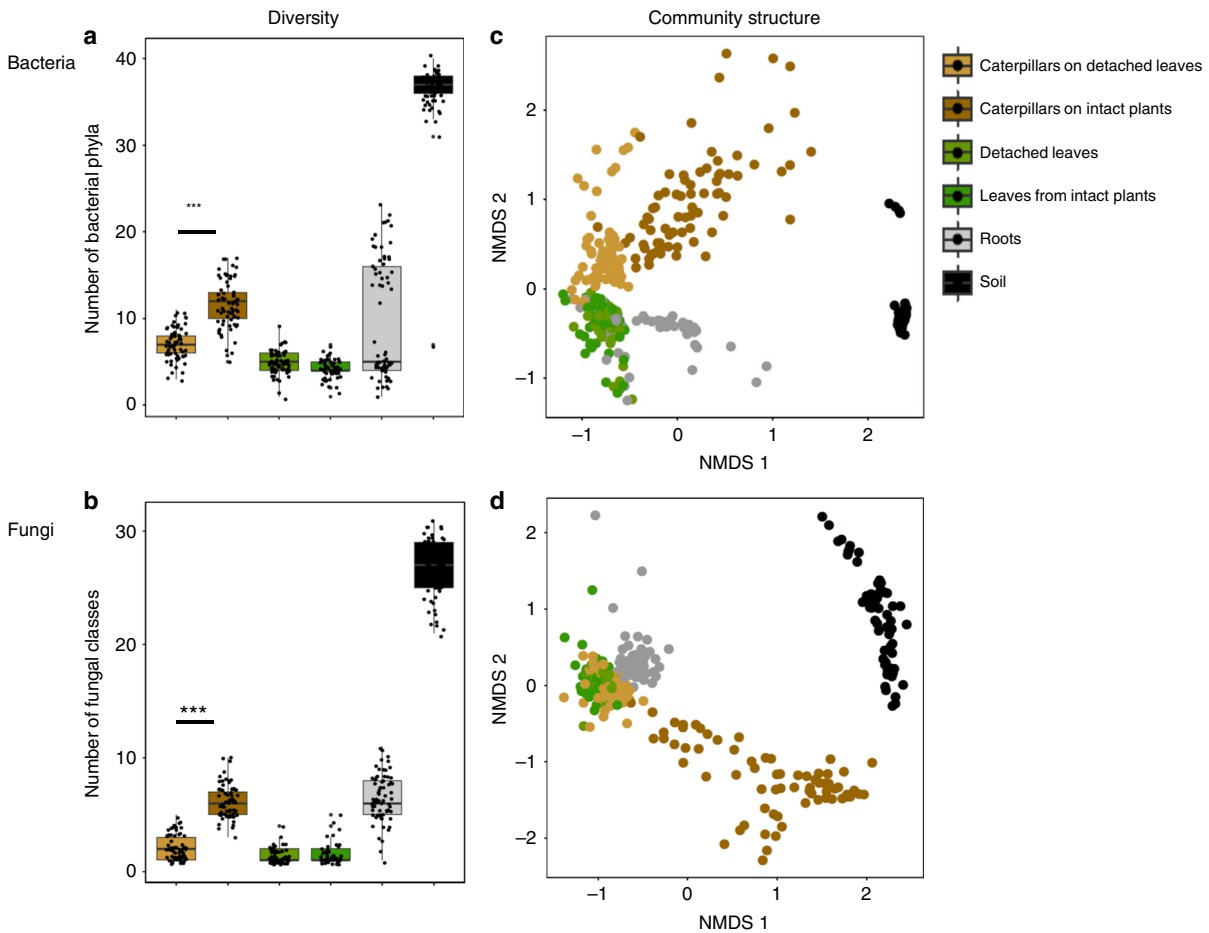

**Fig. 1** Diversity and community structure of bacteria and fungi in caterpillars, leaves, roots and soil. **a** number of bacterial phyla and **b** number of fungal classes of caterpillar, leaf, root and soil samples. Caterpillars were kept on intact plants or on detached leaves. The Tukey box-and-whisker-plots depict median number of phyla and classes in each compartment and variation is shown in the scatter. The raw (Chao1) diversity data is presented in Supplementary Fig. 2, and phyla and their relative abundance in Supplementary Fig. 3 (bacteria) and Supplementary Fig. 4 (fungi). Asterisks (***) indicate significant differences of GLM at the level of $p < 0.001$. **c, d** Non-metric multidimensional scaling (NMDS) of bacterial (**c**) and fungal (**d**) communities. The clustering is based on Bray-Curtis similarity and the resulting 2D stress for the best solution is 0.16 (bacteria) and 0.19 (fungi). Source data for **a** and **b** are provided in a Source Data file

**Soil legacy effects on soil, plant, and insect microbiomes**. We investigated the legacy effects created by field-grown plant communities, on the composition of microbial communities in soils, dandelions grown in those soils, and caterpillars reared on these plants, in two parallel assays (Supplementary Fig. 1). The composition of the plant community (fast- and slow-growing grasses or forbs) that conditioned the soils that were used, influenced the fungal and bacterial community structure in these soils (Fig. 3a, e). Surprisingly, this did not alter the root- or leaf -associated microbiomes in the dandelion plants that were growing in these soils (Fig. 3c, d, g, h). However, we did detect these soil-derived plant community effects in caterpillar microbiomes, but only when the caterpillars were fed on intact plants (Fig. 3b, f), suggesting that, even though they are plant feeders, the caterpillars had been in direct contact with the soil. In the caterpillars fed on intact plants the fungal class Eurotiomycetes and the bacterial phyla Bacteroidetes, Alphaproteobacteria and Betaproteobacteria were significantly affected by characteristics of the plant community that had conditioned the soil (Supplementary Fig. 8).

**Plant and insect biomass and abiotic soil characteristics**. Shoot and root biomass of the test plants were on average higher in soils of fast-growing grass communities, but lower in soils of slow-

growing grass communities than in other soils, both in test plants of the intact plant assay (Supplementary Fig. 9A, C) and of the detached leaf assay (Supplementary Fig. 9B, D). Caterpillar biomass was highest in soils of fast-growing forb communities, and lowest in soils of slow-growing forb communities but only when caterpillars were fed on intact plants (Supplementary Fig. 10). Soil chemical parameters did not differ between soils, except that nitrogen availability was higher in soils from grass communities than in other soils (Supplementary Fig. 11, Supplementary Table 1). There was no relationship between caterpillar biomass and plant biomass, and plant, and caterpillar performance did not correlate with soil chemical parameters (Supplementary Fig. 12). We further related the abundances of fungal classes and bacterial orders in the caterpillars to the performance of the caterpillars. There was a negative relationship between the biomass of caterpillars that were kept on intact plants and the relative abundance of the fungal classes Chaetotyriales, and between the number of surviving caterpillars and the relative abundance of Sordariales, Pseudomonadales and Burkholderiales. Caterpillar biomass and survival were positively correlated with two fungal classes and three bacterial orders (Fig. 4). For the caterpillars that were fed detached leaves, there were no significant correlations between caterpillar biomass and the relative abundance of any fungal orders or bacterial classes (Fig. 4).

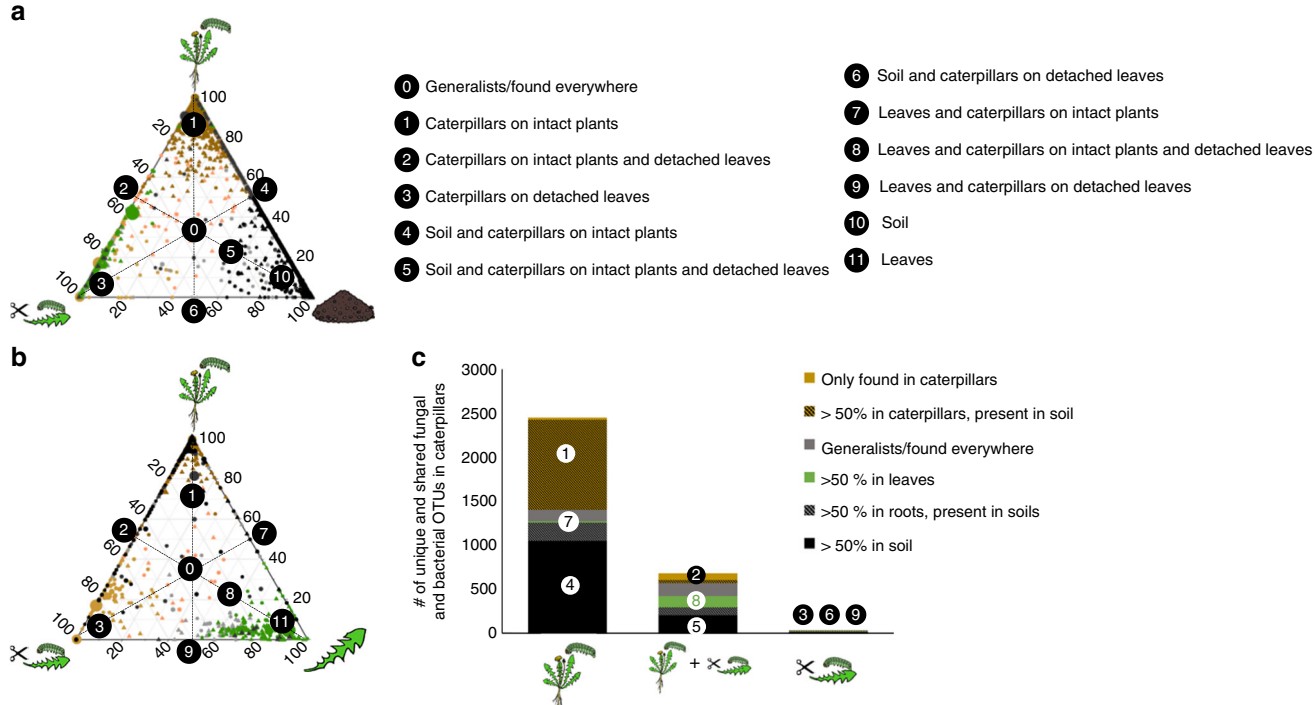

**Fig. 2** Bacterial and fungal OTUs shared among caterpillars, plants and soil. **a, b** Ternary plots of OTUs found in caterpillars. Each symbol represents a single OTU; circles represent bacterial OTUs and triangles fungal OTUs. Only OTUs found in at least 10% of the samples are included in the figure. The size of each symbol represents its relative abundance (weighted average) and its color the compartment where it is primary found. Green depicts OTUs found >50% in leaves, brown depicts OTUs found >50% in caterpillars (dark brown OTUs in caterpillars on intact plants and light brown on detached leaves), black depicts OTUs found >50% in soil, grey OTUs found >50% in roots. Grey symbols represent general OTUs found in all compartments. The position of each symbol represents the contribution of the indicated compartments to the total relative abundance. The 50% lines are drawn in the figure and most important compartments are marked with numbers (0–9). **a** Depicts OTUs shared between soil (right side), caterpillars on intact plants (top) and caterpillars on detached leaves (left) and **b** depicts OTUs shared between plants (right), caterpillars on intact plants (top) and caterpillars on detached leaves (left). **c** The total number of unique and shared OTUs of caterpillars on intact plants and caterpillars on detached leaves. Both fungi and bacteria are included in the figure and their identity on the phylum/class level is shown in Supplementary Fig. 6. The color of the compartment where the OTUs are predominantly found and the corresponding region in panel **a** and **b** is also shown

## Discussion

In this study, we tested the hypothesis that plants would acquire a subset of their phytobiome from the soil and that this would subsequently shape the microbiome of a plant-associated caterpillar. Remarkably, our results show that aboveground caterpillars acquire a large part of their microbiome, not from the plant they are feeding on, but directly from the soil. Over the past two decades a large number of studies have reported that soil microbiota can influence the performance of aboveground plant-feeding insects[12,13,28], but this has been solely attributed to systemic chemical changes in the host plant[29,30]. We now argue that these belowground-aboveground effects may be partly due to direct interactions between insects and soil microbiomes.

Previous studies have already shown that insects can selectively acquire symbiotic bacteria from the genus *Burkholderia* from the soil[19,20,31]. Our results now show that entire microbiomes of caterpillars on intact plants are affected by soils, and that they are enriched in particular bacterial and fungal genera, disproportionate to their relative presence in soils. When the caterpillars were fed detached leaves, this was not observed. Both Eurotiomycetes and Actinobacteria, the genera found disproportionally more in the caterpillars on intact plants than in soils and in caterpillars fed detached leaves, are known to act as insect symbionts and produce antibiotic compounds[15,32,33]. Furthermore, caterpillars that were in contact with soils had acquired species of yeasts commonly found in soils but that have recently been identified as symbionts of insects[34] and found in

large numbers in human guts[35]. This suggests that leaf eating insects may actively acquire more species of beneficial microbes from the soil than what is known from literature so far[19]. However, we observed both positive and negative relationships between the relative abundance of soil microorganisms and the performance of the caterpillars, indicating that the acquisition of microbes from the soil by insects may not always be beneficial. Recent work indicates that caterpillar microbiomes may be transient[10]. Our findings that soils shape insect microbiomes now offer a viable explanation why these microbiomes are variable even within a single insect species. Caterpillar microbiomes reflect their (soil) environment and as soil microbiomes vary temporally and spatially[36], this may also affect the microbiomes of the caterpillar. An important question that remains to be answered is how persistent these soil effects on insect microbiomes are and to what extent they change when insects encounter new soil microbiomes as they move or grow.

Remarkably, our results also show a link between the composition of the plants that previously grew in the soil and insect microbiomes. The consequences of (microbial) soil legacy effects for plant growth and plant-insect interactions have received considerable attention recently[25,37]. Our study now shows, for the first time, that such soil legacy effects can influence the performance of aboveground insects as well as their microbiomes. However, interestingly, these legacy effects on caterpillar performance and insect microbiomes were only observed in caterpillars that were fed on intact plants, and not when they were fed on

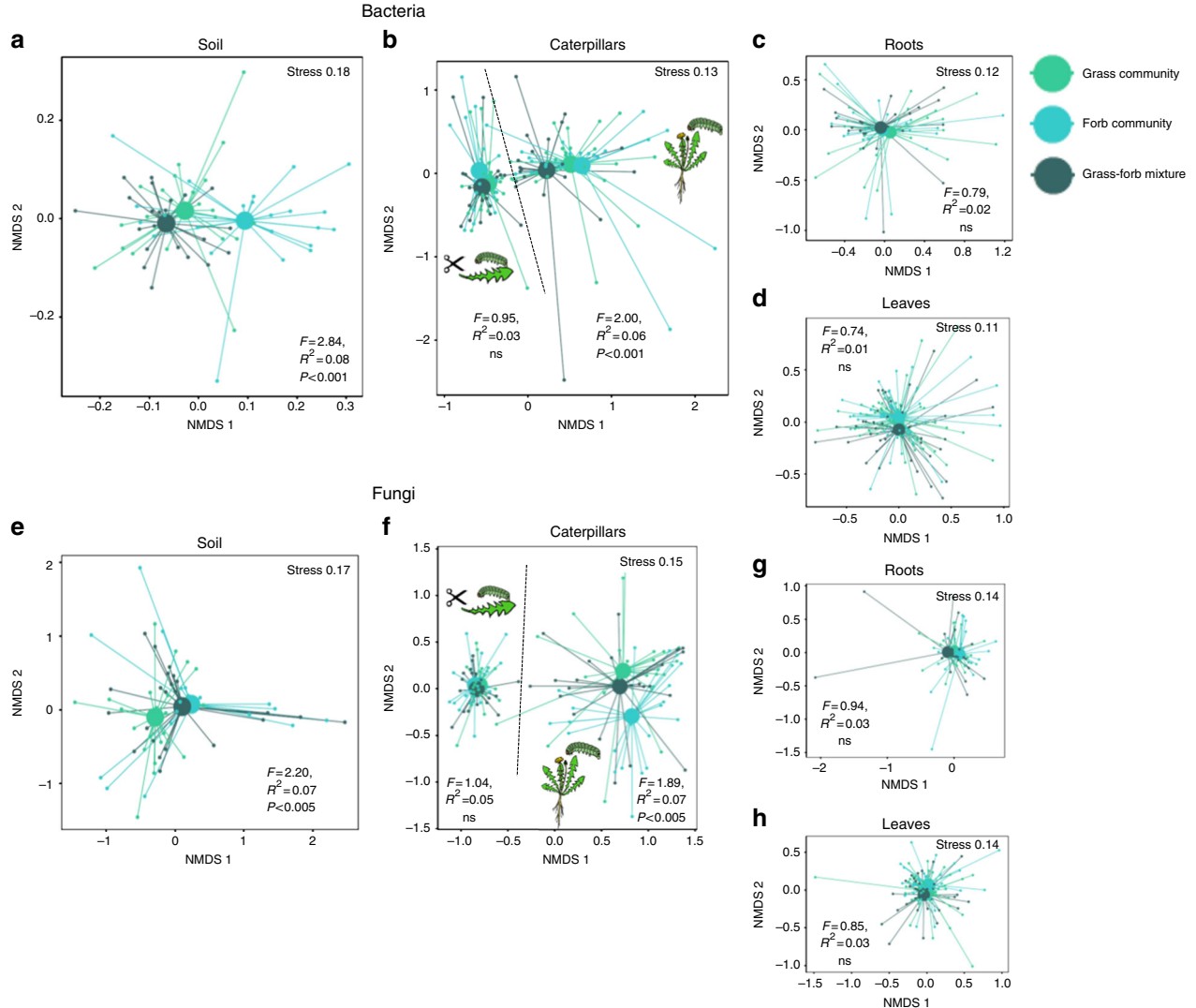

**Fig. 3** Legacy effects of plant communities on microbiomes. Plant community identity effects on bacterial **a**–**d** and fungal (**e**–**h**) communities in caterpillars, leaves, roots, and soil. NMDS plots are presented based on Bray–Curtis similarity. The 2D stress value for each panel ranges between 0.11–0.18. Soils originating from grass communities are presented with light green symbols, soils from forb communities with turquoise symbols and soils from mixed grass and forb communities with dark green symbols. In each panel, smaller symbols depict individual samples, centroids are depicted with larger markers. Significance of the plant community treatment effect based on a PERMANOVA is also presented in each panel. **a**, **e** represent the composition of microbiomes in soils, **b**, **f** microbiomes in caterpillars both on intact plants and on detached leaves. **c**, **g** microbiomes in roots and **d**, **h** microbiomes in leaves. The effect of plant community growth rate (fast- and slow-growing communities) is shown in Supplementary Fig. 14

detached leaves. This is important, as it suggests that soil legacies may not only influence insects mediated via plant quality, but that there may be a direct link between soils and insects, via the microbiome.

It is important to note that the test plant and insect microbiomes were investigated under artificial conditions in the greenhouse. Under natural conditions, insects may acquire a higher proportion of their microbiomes from dietary sources than we observed in this study. For instance, leaf microbiomes of host plants may be enriched by environmental microbiomes, e.g. via rain splash or wind[38]. As such, in natural settings, the dynamics of microbiome acquisition may vary from those observed in this study. Polyphagous caterpillars, such as the one used in this study, can often be found on soil e.g. because they move up and down the plant and regularly change host plants[25]. Hence they may also have more frequent contact with the soil under natural conditions than in the artificial greenhouse setting with individually potted plants that we used in this experiment.

A potential caveat in our study is that instead of a bottom-up pathway, the caterpillar microbiomes may have caused changes in the composition of the soil or leaf microbiomes e.g. excreted via their frass. However, we consider this unlikely for two reasons. First, there were no differences in microbial composition between the leaves that were in contact with caterpillars (and their frass) and leaves from the plants which had no insects. Second, insects weighed only 15 mg at the end of the experiment and the amount of frass produced by these small insects was marginal relative to the amount of soil used in each pot. However, studies with soil and insect microbes, labeled with isotopic tracers should further examine the direct and indirect interactions between soil, plant and insect microbiomes. Future studies should also address the functional consequences of soil legacy effects on microbiomes of aboveground insects and how widespread this phenomenon is among insect taxa.

A second caveat is that differences in size of the caterpillars in the two parallel assays may have contributed to the observed

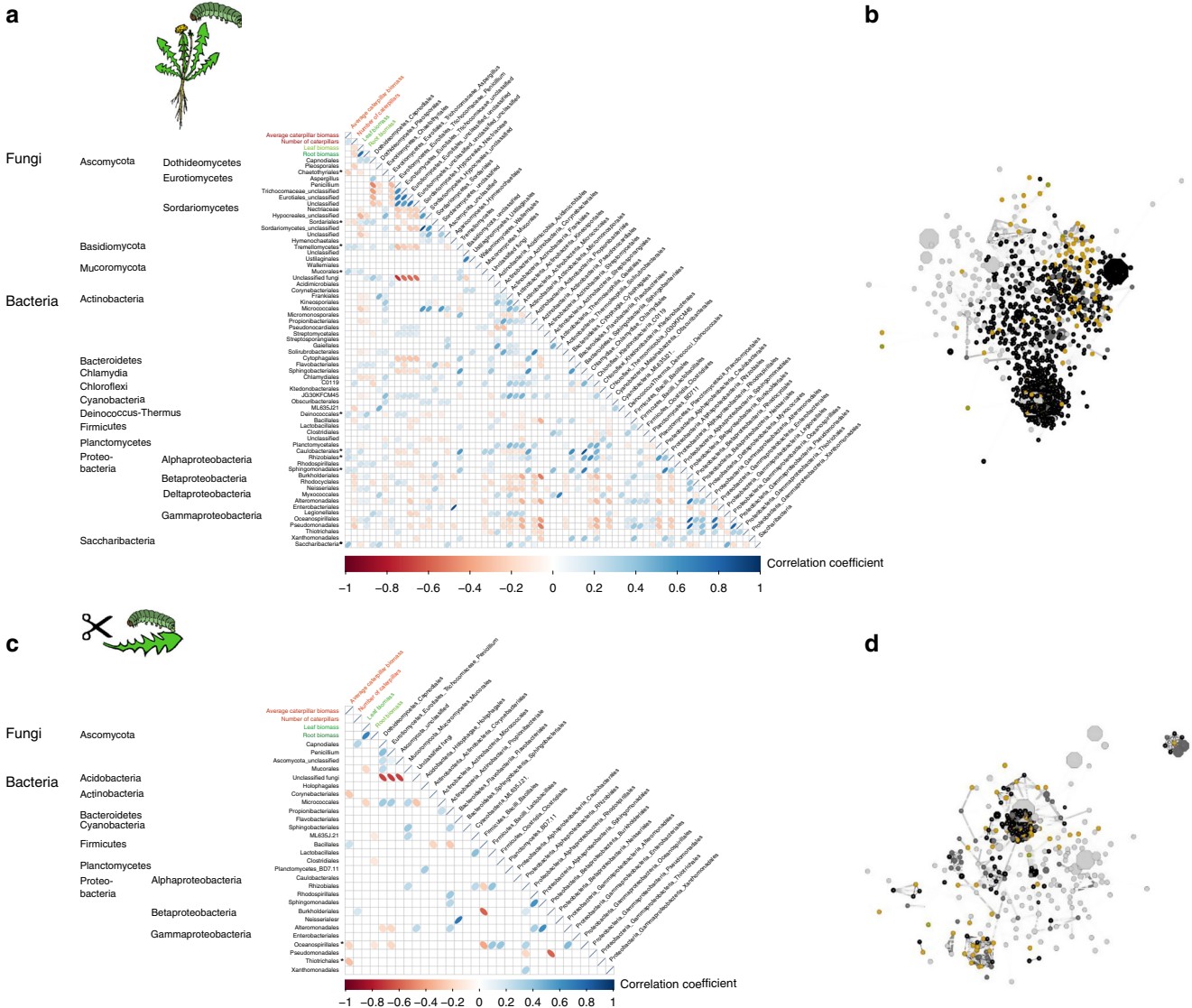

**Fig. 4** Correlations between caterpillar parameters, plant parameters, and relative abundance of fungal and bacterial taxa in the caterpillars. **a** fungal orders and bacterial classes detected in caterpillars fed on intact plants, and **c** on detached leaves. Correlations are based on linear Pearson correlation coefficients against each other and average caterpillar biomass (red), caterpillar survival (red), and leaf- and root biomass (green). The scale color of the filled squares indicates the strength of the correlation (*r*) and whether it is negative (red) or positive (blue). All correlations are corrected with FDR and only significant correlations with *p* < 0.05 are shown. If the correlation is not significant, the box is left white. Asterisks next to names of taxa mark significant correlation between this taxon and caterpillar performance. **b** and **d** represent a network of all significant co-occurrences (Spearman rank correlation coefficient with Bonferroni correction, *p* < 0.01) of OTUs in caterpillars on intact plants (**b**) or on detached leaves (**d**). The size of the nodes represents the relative abundance of the OTUs (weighted average) and the color represents the compartment where it is primary found. Green depicts OTUs found mostly in leaves, brown OTUs in caterpillars (dark brown OTUs of caterpillars on intact plants and light brown OTUs of caterpillars on detached leaves), black depicts OTUs found primarily in the soil and grey OTUs that are general in all compartments

differences in caterpillar microbiomes. In the detached leaf assay, caterpillars were reared to L3 stage, until there were no more suitable leaves available on the source plants. At this point, the caterpillars in the parallel intact plant assay were considerably smaller (L2). As it is known that insect microbiomes differ between larval stages[9,31,39], the intact plant assay was continued until the caterpillars had molted to L3. Although the caterpillars were bigger on whole plants than on detached leaves (Supplementary Fig. 13) when they were collected, their average biomass differed only by 4.4 mg. *M. brassicae* is known to grow well over 200 mg on various plant species that grow in similar soil types[25]. Therefore, it is unlikely that these differences are the main driver of the observed differences in microbiomes. The small size of the caterpillars did not allow for proper removal of the gut, which is

the reason why we extracted caterpillar-associated microbiomes from whole caterpillars[14]. However, we used generally accepted methods in microbial ecology to sterilize surfaces[3] to thoroughly clean the insect cuticle. We detected various cuticle-associated insect pathogens in the soils, which also correlated negatively with insect performance, but we did not observe these pathogens in the insect samples, suggesting that our sterilization procedure was effective in eradicating cuticle-bound microbes and thus that it likely reflects the internal insect microbiome.

We conclude that soil and insect microbiomes are linked, but that this is not mediated by the host plant, and that the role of soil microbiomes in modulating aboveground food-webs should be re-evaluated. Until now this has been overlooked, and the current results stress that studies on the composition and functioning of

the microbiomes of plant-feeding insects should be carried out under conditions in which insects have access to the soil and soil microbiome that the host plant is growing in. Finally, an increasing number of studies is now showing that insect microbiomes may be important for insect fitness. We stress that these insect microbiomes can be the consequence of legacy effects of previous generations of plants on soil microbiomes.

## Methods

**Field design and soil sampling**. To create specific soil legacies, field plots were set-up in an existing grassland in the nature area De Mossel (N 52° 3′, E 5° 44′, Natuurmonumenten, Ede, The Netherlands). Each field plot measured 80 × 250 cm, and between plots there were 1-m-wide paths that were mown regularly. In May 2015, the vegetation (sods) of each plot was removed at 4 cm depth to remove the majority of the roots. The plots were subsequently sown with fast- and slow-growing grass and forb species that are common in this grassland ecosystem. Each plot was sown with three grass species, three forb species, or with a mixture of three grass and three forb species. The total seed density in each plot was 12450 seeds, equally divided over the species in the community. There were three different fast- and three different slow-growing grass, forb and mixed communities (totalling 18 communities, see table S2 and S3) and there were four replicate plots for each community (72 plots in total). To maintain the composition of the sown communities, plots were hand-weeded regularly in 2015 and 2016.

In February 2017, live field soil was collected from each plot from the top 10 cm of the soil, as most of the roots are concentrated in this top layer[40]. Soils were sieved to remove roots, stones and most macro-invertebrates (sieve mesh Ø1.0 cm). Live soils were then mixed with sterilized bulk field soil (1:2 live:sterile v/v). Sterilized soil was obtained by γ-irradiation (>25 Kgray, Synergy Health, Ede, The Netherlands), of homogenized soil that was collected from the same field site. 11 × 11 cm square pots were filled with 1000 g of mixed soil. Two pots were filled with the same soil for each of the replicates in this experiment. A priori, one of the two pots was assigned to the detached-leaf assay while the other was assigned to the intact-plant assay. There were 18 plant community-conditioned soils, four independent field plot replicates, and two types of bioassay resulting in a total of 144 pots (Supplementary Fig. 1A, B). After filling, pots were acclimatized in a climate controlled greenhouse (light regime 16:8, L:D, day temperature 21 °C, night temperature 16 °C, relative humidity 50%) for 1 week, allowing the soil microbial communities to recover.

**Test plants**. Common dandelion (*Taraxacum officinale*, Asteraceae) was used as a model species. Dandelion is a perennial lactiferous plant with a broad geographical distribution that occurs in most of the temperate and subtropical regions of the world[41]. Several recent studies have used dandelion to address various ecological questions[42,43]. In this study, seeds of *T. officinale* were genetically identical, as they were obtained from a single clonal (apomictic) maternal line. Before germination, seeds were surface-sterilized using 2.0% bleach solution and then thoroughly rinsed with demineralized water. Seeds were geminated on sterile glass beads in a climate cabinet (light regime 16:8, L:D, day temperature 21 °C, night temperature 16 °C).

We transplanted one *T. officinale* seedling per pot when the seedlings were one-week-old. Dandelion leaves grow upwards in pots and thus, the rosettes are not in direct contact with the soil (Supplementary Fig. 1C). Pots were randomly distributed in the greenhouse and plants were grown for five weeks under controlled conditions (light regime 16:8, L:D, day temperature 21 ± 1 °C, night temperature 16 ± 1 °C, relative humidity 50%). The plants were watered with demineralized water three times per week to keep a constant soil moisture level. Each plant received 60 ml of 50% diluted Hoagland (1:1 Hoagland:demineralized water, v/v) nutrient solution in week 3 and 4, to mitigate the effects of nutrient limitation. The plants were used for assays when they were five weeks old.

**Insect-plant assays**. Eggs of the polyphagous cabbage moth, *Mamestra brassicae* (Lepidoptera: Noctuidae) were obtained from the Department of Entomology at Wageningen University, The Netherlands. The larvae were originally collected from organic cabbage fields near the university. The cabbage moth had been mass-reared for several generations on Brussels Sprouts, *Brassica oleracea* var. *gemmifera* cv. Cyrus. The eggs laid by a cohort of females were surface-sterilized using 2.0% bleach solution and rinsed with demineralized water and then dried with sterile filter paper. The eggs were subsequently transferred to sterile petri-dishes and kept in a climate cabinet (light regime 16:8, L:D, temperature 21 °C). Upon hatching, *M. brasciae* larvae were fed on artificial diet (Supplementary Table 4) until they reached the second larval instar stage.

We tested the effects of each of the soils on *M. brassicae* caterpillars in two parallel assays in order to disentangle the plant-mediated and the direct soil effects on caterpillar microbiomes. The outline of these two assays is shown in Supplementary Fig. 1D. The assays were performed parallel to each other and we used second instar *M. brassicae* larvae, randomly selected from several hundred mass-reared larvae which were grown under sterile conditions. In one assay, caterpillars were fed with leaves clipped from plants that were growing in the different soils, and in the other they were fed on intact caged plants growing

in soil from the same origin. For the first assay we cut the largest fully expanded leaf of each plant using sterile curved razor blades and placed it on a sterile petri-dish with the petiole covered with a piece of wet cotton that was soaked in demineralized water to prevent dehydration during the assay. Five *M. brassicae* caterpillars were placed in each petri-dish that contained one detached-leaf. After ± 24 h, the leaf was removed and replaced by a newly collected leaf originating from the same plant. We conducted the detached-leaf assay for 5 days due to the limited availability of suitable leaves after which the caterpillars were collected and their biomass was measured. Caterpillars from this experiment were collected to be used for molecular analysis. In the second assay, *T. officinale* plants were transferred individually to fine-meshed (300 μm) polyester sleeves and five *M. brassicae* larvae were placed on each individual plant. As growth of the caterpillars was much faster on the detached leaves (which we may speculate to be due to the absence of herbivore-induced defences in these plants[44]) and caterpillar microbiomes are known to differ between larval stages[45], we kept the insects on the plant until they were of the same larval stage (L3) and visually similar in size (Supplementary Fig. 13). Thus, in the intact-plant assay the caterpillars were allowed to feed and move freely on the plant for 14 days. Caterpillar mortality was recorded and fresh biomass of each individual caterpillar was measured and averaged per cage. Shoot and root biomass was collected after the insects were removed from the plants and dry weight was measured after oven drying (60 °C for 4 days).

**Soil, plant, and caterpillar sampling for microbiome analysis**. We collected samples of surface-sterilized caterpillars, and leaves for analysis of the microbiomes[3] from both assays. Leaves were collected from three leaf discs from each of three individual fully expanded leaves using a sterile 25 mm sample puncher. In the intact plant-assay leaves with clear signs of caterpillar feeding damage were selected for the analysis. Leaves for the detached leaves were selected from the corresponding plants at the same time point. The leaf discs were flash-frozen in liquid nitrogen and then stored at −80 °C until processing.

From the intact plant assay we further collected and surface-sterilized roots and rhizosphere soil. All caterpillar and root samples were surface-sterilized by dipping them in 2.0% bleach for 30 sec and then rinsed with autoclaved demineralized water. The caterpillars and roots were subsequently transferred to a new 15 mL falcon tube filled with 10 mL autoclaved Dulbecco's phosphate buffered saline (DPBS, Sigma-Aldrich, Darmstadt, Germany) and then sonicated in a BRANSONIC ultrasonic cleaner (Bransonic ultrasonics, Danbury, USA) for 10 min (ten cycles of 30s ultrasonic burst, followed by 30s rest) in order to disrupt microbes that were attached to the exterior surfaces[3]. After sonication, the caterpillars and roots were rinsed with autoclaved demineralized water three times and then stored at −80 °C until processing. Leaf, root and caterpillar samples were lyophilized prior to DNA extractions. Rhizosphere soils were collected from the intact-plant assay by first removing the bulk soil by shaking the root system and then gently removing the remaining soil above a sterile tray. This soil was stored in -80°C until processing.

**Soil chemical analysis**. For soil chemistry measurements, the soil samples were air dried at 40 °C and sieved through a 2 mm sieve. For extraction, 3 g dry soil was combined with 30 ml of 0.01 M $CaCl_2$ and shaken for 2 h at 250 rpm. After centrifugation at 3000 rpm for five minutes, 15 mL of the supernatant was filtered through a syringe filter with cellulose acetate membrane. Then 12.87 mL of filtrate and 130 μL $HNO_3$ were vortexed and extractable elements (Fe, K, Mg, P, S, and Zn) were measured the next day (ICP-OES, Thermo Scientific iCAP 6500 Duo). The remaining part of the filtrate was used to measure pH, and measure $NO_2 + NO_3$ and $NH_4$ on a QuAAtro Autoanalyzer (Seal analytical).

**Molecular analysis of soils, plants, and caterpillars**. For root, leaf and caterpillar samples, bead beating and DNA extraction were performed with the MP Biomedical FastDNA™ Spin Kit. For the soil samples, DNA was extracted using Qiagen DNeasy PowerSoil Kit. Approximately 10 ng of template DNA was used for PCR using primers ITS4ngs and ITS3mix targeting the ITS2 region of fungi[46]. For bacteria we used primers 515FB and 806RB[47] targeting the V4 region of the 16 Sr RNA gene. Presence of PCR product was checked using agarose gel electrophoresis. The PCR products were purified using Agencourt AMPure XP magnetic beads (Beckman Coulter). Adapters and barcodes were added to samples using Nextera XT DNA library preparation kit sets A-C (Illumina, San Diego, CA, USA). The final PCR product was purified again with AMPure beads, verified using agarose gel electrophoresis and quantified with a Nanodrop spectrophotometer before equimolar pooling. Separate libraries were constructed for bacteria and fungi, and from rhizosphere soil samples (72 samples per library) and a combination of samples derived from leaves, caterpillars of the plants allocated to the detached leaf and intact plant bioassays, and roots (360 samples). This made the total data collected to be 4 runs on a MiSeq. Libraries were sequenced at McGill University and Genome Quebec Innovation Center. For all compartments, extraction negatives were used and further sequenced. A mock community, containing 10 fungal species, was included to compare between sequencing runs and to investigate the accuracy of the bioinformatics analysis.

**Bioinformatic and statistical analysis**. The bacteria data were analysed using an in-house pipeline[48] using the SILVA database with SINA classifier. The PIPITS pipeline[49] was used to classify fungi. Taxonomy was assigned using the rdp classifier against the UNITE fungal ITS database[50]. Finally, the OTU table was parsed against the FunGuild (v1.1) database to assign putative life strategies to taxonomically defined OTUs[51]. All singletons and all reads from other than bacterial or fungal origin (i.e. plant material, mitochondria, chloroplasts and protists) were removed from the dataset. The resulting data included approximately 10 million good quality (QC over 28, overlap over 25 bp, length over 100 bp, no chimeras) paired sequences for bacteria and 7.9 million sequences for fungi.

Samples that had over three times lower or higher number of reads than average in the same compartment were removed from the dataset. This resulted in removal of 1–10 samples out of 72 depending on organisms and compartment (Table S5). Furthermore, sequence count in a sample was used as a co-variate in the model when Chao1 and relative abundances of fungal classes and bacterial phyla were analysed to prevent the sequencing depth having effect on the results. Data was normalized using the cumulative sum scaling (CSS) after exploring several other normalization options[52]. We used the Adonis function with Bray-Curtis dissimilarity (permutational MANOVA using distance matrices; R package Vegan[53]) to test whether microbial composition differed between sample types and plant community legacies, including species identity as an explanatory variable and the matrix of community dissimilarities among samples as the response. Separations among treatments were visualized using non-metric multidimensional scaling (NMDS) of a Bray-Curtis dissimilarity matrix using square transformation and Wisconsin standardization. For the OTU level analysis, the presence of each OTU in each compartment was individually calculated. As a rule, for an OTU to be present in a compartment, it needed to be present in more than 10% of the samples of the compartment. The ternary plots were created using package ggtern[54]. Generalized linear models (GLM) were used to compare the diversity and Chao1 index and the relative and absolute abundances (counts) of bacterial phyla and fungal classes between compartments and legacies. The Chao1 data was ln transformed prior to analysis to fulfil the requirements of normality. Sequence count was used as a co-variate in the analysis. To account for the overdispersion in the model when comparing different compartments, we used Poisson distribution in our generalized linear model (GLM) for the count data. Further, we fitted zero-inflated Poisson regression models (package PSCL in R) but with our data they were not superior to GLM with Poisson (Vuong test; $P > 0.05$). The results of GLM were evaluated with a Chi-square test and a Tukey post-hoc test. To analyze the effects of different soil legacies on bacterial and fungal taxa and on caterpillar biomass, linear mixed effects models (LME) were used from the package nlme as the data within each compartment were generally normally distributed. All $p$-values derived from multiple calculations were corrected with Benjamini & Hochenberg which relies on calculating the expected proportion of false discoveries among rejected hypotheses to control for false discovery rate (FDR)[55]. All numerical data were checked for (multivariate) normality and log-transformed if necessary. To create networks the co-occurrence of each OTU present in more than 10% of the samples of the caterpillars was calculated using Spearman rank correlation coefficients following a Bonferroni correction ($P < 0.05$) as a cut off for a significant correlation between two OTUs[56]. The networks were visualised in Cytoscape[57]. All statistical analyses were performed in R version 3.4.4[58].

**Reporting summary**. Further information on experimental design is available in the Nature Research Reporting Summary linked to this article.

## Data availability

Paired-end DNA sequencing reads for this project have been deposited in the European Nucleotide Archive under accession number PRJEB27512 [https://www.ebi.ac.uk/ena/data/view/PRJEB27512]. Plant and caterpillar growth data and soil chemistry data are deposited in Dryad [https://doi.org/10.5061/dryad.99504fd].

## Code availability

Custom code used for the analyses that support this work is available in R upon request.

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

## Acknowledgements

We thank Wim van der Putten and Jos Raaijmakers for commenting on an earlier version of the manuscript, Luuk Wilbers for assistance, and Koen Verhoeven for providing seeds of the clonal *T. officinale* line. Sequencing of the samples was performed in collaboration with McGill University and Génome Québec Innovation Centre, Canada. This work was funded by the Netherlands Organization for Scientific Research (NWO VICI grant 865.14.006). NIOO-KNAW publication nr 6682.

## Author contributions

F.Z. and T.M.B. conceived the idea of the experiment. F.Z., R.H., and T.M.B. optimized the experimental design. F.Z. and R.H. performed the greenhouse experiment. F.Z., R.H., and S.E.H. performed the molecular work. S.E.H. performed the bioinformatic and data analyses. S.E.H., F.Z., R.H., and T.M.B. contributed equally to writing the manuscript.

## Additional information

**Competing interests:** The authors declare no competing interests.

