## [Peer Review File · Nature Communications]

Reviewers' Comments:

Reviewer #1:

Remarks to the Author:

In this study Hannula and co-authors explore how soil and insect herbivore microbiomes may be linked. To explore this question the authors alter soil microbiomes by rearing different types of plant communities to condition these soils, and then rear a moth on a plant species growing in these different soil types. The authors compare microbial communities in different compartments including soil, leaves, insects and roots. To assess from where insect microbiomes originate, the authors also rear a group of larvae in detached leaves so that these insects are not in direct contact with the soil. The topic of the study is exciting and the paper is well written, the authors present an impressive amount of data. Despite these merits I have few important remarks that I believe the authors should address before publishing this study in Nature Communications.

1. After reading the manuscript a couple of times I miss a clear positioning about the hypotheses tested in the study. I can easily picture them, but they are not specifically presented in the last bit of the introductory section. For example, although it may seem obvious, do the authors hypothesise that similar microbiomes in the soil and in caterpillars feeding on entire plants (not in detached leaves) imply that the latter community comes from the former?

2. Most of the conclusions of this study are based on comparisons among microbiome composition in the soil and in caterpillars reared either directly on the plant (probably in close contact with soil) or on detached leaves. Microbiomes of caterpillars feeding on entire plants are more similar to those of the soil than to microbiomes of caterpillars feeding on detached leaves. Microbiomes of caterpillars on entire plants are also more diverse than those of caterpillars on detached leaves. Based on these results the authors conclude that caterpillar microbiomes originate from the soil. I have two important issues with these results. The first one is that I am not sure the microbiome of caterpillars feeding on entire plants and on detached leaves are easily comparable. Larvae in entire plants were allowed to feed on plants during 14 days, but only 5 days on detached leaves. How can the authors rule out that observed differences are not just natural changes occurring along insect development? This has been found, for example, in *Spodoptera littoralis*: Chen, B. et al - 2016 - Biodiversity and activity of the gut microbiota across the life history of the insect herbivore *Spodoptera littoralis*. *Sci. Rep.* My second concern is that if caterpillars were allowed to freely move and feed on plants for 14 days, isn't it possible that the similarity between caterpillar and soil microbiomes are due to caterpillars "contaminating" the soil with their faeces? Moth's guts are quite alkaline and a hostile environment for microbes so it is possible that this explanation is more likely than what the authors propose. This could explain, for example, why some microbial taxa that are present in the soil are highly abundant in caterpillars. This issue would have been easily solved by feeding caterpillars on entire plants inside clip cages (not sure this is possible as I am not familiar with the biology of this moth). Alternatively, a soil control with plants, but without caterpillars (maybe imposing mechanical damage to plants to emulate herbivory) should be added.

3. Please see these references as insect herbivores have already been reported to acquire symbionts from the soil. Kikuchi Y, Hosokawa T, Fukatsu T (2011) Specific developmental window for establishment of an insect-microbe gut symbiosis. *Appl Environ Microbiol* 77:4075–4081. Kikuchi Y, Hayatsu M, Hosokawa T et al (2012) Symbiont-mediated insecticide resistance. *Proc Natl Acad Sci*

USA 109:8618-8622

Enric Frago - CIRAD

Reviewer #2:

Remarks to the Author:

This research focuses on how plant community alteration of the soil microbiome influences the establishment of herbivore associated microbial communities. Soil from field plots was brought into the greenhouse to serve as a substrate to grow a common plant (dandelions). Herbivores (*Mamestra brassicae* caterpillars) were placed on these plants or on cut leaves from these plants. The researchers conducted an exhaustive set analyses of the relationships between the bacterial and fungal communities in the soil, on and in the plant leaves, and on and in the caterpillars. Their main finding is that caterpillars fed on intact plants are associated with a more diverse microbial community that is more similar to that of the soil than the caterpillars fed on cut leaves disassociated from the soil community. They interpret this finding as suggesting that caterpillar acquisition of microbes from the soil is an underrecognized component of how these herbivores establish their microbiomes. They further find that plant community composition alters the microbiome of soil, which in turns alters the microbiome of caterpillars reared in the presence of this soil.

There is a significant problem with the experimental design that makes me wary of the authors' main conclusion, namely that caterpillars fed on intact plants have a more diverse gut microbiome because they have access to the soil microbiome. That concern is that the caterpillars in the intact plant treatment were allowed to feed on plants for 14 days, *ad libitum*. The caterpillars in the cut leaves treatment were only fed on the plant leaves for 5 days, with the leaves needing to be changed every day. Therefore, the caterpillars in two treatments are different ages and likely different life stages. Furthermore, based on the descriptions, it is not clear if the cut leaf treatment caterpillars ran out of food some days, which would undoubtedly cause metabolic changes and stress signals that could alter the gut microbiome.

I am also concerned that larvae were fed on artificial diet until second instar, precluding establishment of the natural gut microbiome in the stage likely when most caterpillars would begin to establish this microbiome. This is particularly of concern if this diet contained antibiotics. The artificial diet recipe is not detailed in the methods, so I am not sure if antibiotics are a problem.

I am also concerned that these results must be cautiously interpreted given that the main way that plant leaves acquire foliar endophytes is through rain shower, causing greenhouse grown plants to have few foliar fungal endophytes, for example, compared to outdoor reared plants. Thus, in nature, leaves likely have very different and more diverse microbial communities that likely shape the microbial composition of their insect folivores. Therefore, suggesting the that soil is a more important driver of the insect microbiome in the plant leaves could be true in a greenhouse setting, but may not be true in nature.

A final concern is that the caterpillar samples, as far as I can tell, are whole caterpillars rather than gut samples. While effort was made to surface sterilize these caterpillars, such efforts may not remove all residual DNA. Thus, the increased diversity of the intact plant associated caterpillars could be due to the fact that there were reared in a tent with a whole plant and soil rather than in a sterile Petri dish with a leaf. This could lead to more diverse communities on their exoskeleton, but these may not be functionally relevant. Most previous research on the microbiome of caterpillars has focused instead on the gut microbiome.

This manuscript should be carefully checked for grammar. For example, many sentences are missing commas, and phrases such as "low diverse" should be corrected.

Weight data should be made available in Dryad or another public database.

Supplemental Figure 5. The taxa names in the legend should be simplified.

The abstract does not mention the insect species, a problem given that the title does not either.

Reviewer #3:

Remarks to the Author:

The manuscript by Emillia Hannula et al. seeks to answer which are the forces that drive microbiome composition in plant-feeding insects. The found that caterpillars get most of their microbiome from the soil as opposed to the plants they eat, and that previous history of plants grown in those soils also influence caterpillar microbiome composition. I enjoyed reading the manuscript and I think it presents a series of relevant results from well designed experiments which will be appreciated by a broad scientific audience. However I have some concerns which I list below:

Major comments:

1. My major concern is the lack of any form of multiple testing correction. The authors provide many examples of differential diversities, taxa and correlations with nominally significant p-values, but none of them seem to have been adjusted for multiple testing. This adjustment must be done throughout the manuscript to allow for interpretation of the results.
2. My search for accession number 'PRJEB27512' at the EBI-ENA got no results. Please ensure that the data is made publicly available. Also, it is important that all the relevant sample metadata is made available.
3. The authors present many compelling visualizations and summary statistics showing that the caterpillar microbiota comes from the soil. While they are all consistent with the author's interpretation, they are also indirect. Bray-Curtis focuses on relative abundances differences between shared taxa, and therefore one can easily construct toy examples of NMDS plots like the ones of fig 1 where the OTU sharing is identical between the groups. The ternary plots from fig 2 are also not ideal, since they show relative abundance ratios and not membership. I think a more direct test of the author's claim would be an anlysis of OTU prevalence. For example, how many OTUs are present in almost all intact plant caterpillar and soil samples but never in leaves and detached leaves caterpillars? And what proportion of the community these OTUs make? A clustering and heatmap with presence absence data might also be useful
4. The authors tested alpha-diversity using linear models, however the model specification is not provided. Was it a Gaussian model? Which variables were included? Was any of the variables transformed? More importantly, it is not clear if the authors controlled for potential systematic differences in sampling depth. Cumulative sum scaling was benchmarked only for metrics of beta-diversity, therefore it is important to include terms for sequencing depth in tests for differences of alpha-diversity.
5. The authors tested the abundance of many specific microbial taxa across different treatment groups. The model used was not specified here either so the same clarifications as above are needed here. Moreover, the standard approach would be to use a form of negative binomial model that accounts for sequencing depth differences, overdispersion and that models mean-variance relationships accurately. It appears that the authors performed some type of gaussian ANOVA, which is not really appropriate for count data.

6. In both, test of diversity and microbial abundances, only some comparisons are highlighted with a p-value in the visualizations, but it is not clear which tests were actually performed. Did the authors performed all pairwise comparisons? ANOVA followed by Tukey? The authors should clarify which tests were performed and correct for multiple testing appropriately.
7. I didn't understand if the caterpillars are genetically identical. I guess not. Given the high replication, I don't think this challenges the results but the authors should be explicit about this.
8. One of the main conclusions is that caterpillars get their microbiota from the soil, but little discussion is given as to how this might happen. Is this simply because they occasionally walk on the soil? Or is the transfer mediated by the plant or air? Moreover, I wonder how the authors reconcile this observation with previous work that claims that caterpillar microbiome is mostly transient. how biologically relevant is the caterpillar microbiota for the caterpillars themselves?
9. The authors find that some microbes are associated with caterpillar performance mostly when fed on intact plants growing in conditioned soil. I wonder how likely are the performance benefits due to the microbes themselves versus some abiotic edaphic factor like micronutrient composition, which could be mediated by the plant. Did the authors perform micronutrient analysis in the conditioned soils? Or in the plant tissue? Was there any plant performance difference? And did plant performance correlate with caterpillar performance?

Other comments

1. The resolution of the figures in my review copy was low. Please make sure you have high resolution versions for the actual publication.
2. Colors for soil and root where hard to distinguish in my printed copy. Make sure you pick colors that are easily differentiable.
3. Axis labels are really small in many figures, for example in taxonomic abundance plots.
4. For figure 4, I recommend indicating more clearly in the figure which rows/columns are caterpillar metrics, and which are microbial abundances
5. I like supp figure 8, but it would be better if the diagram also included the creation of different soil legacies with different combinations of plans. It would also be nice if the diagram included the number of replicates of each class.
6. Line 159 of methods: there is a typo 'co-occurrence' should be 'co-occurence'

Reviewers' comments:

Reviewer #1 (Remarks to the Author):

In this study Hannula and co-authors explore how soil and insect herbivore microbiomes may be linked. To explore this question the authors alter soil microbiomes by rearing different types of plant communities to condition these soils, and then rear a moth on a plant species growing in these different soil types. The authors compare microbial communities in different compartments including soil, leaves, insects and roots. To assess from where insect microbiomes originate, the authors also rear a group of larvae in detached leaves so that these insects are not in direct contact with the soil. The topic of the study is exciting and the paper is well written, the authors present an impressive amount of data. Despite these merits I have few important remarks that I believe the authors should address before publishing this study in Nature Communications.

1. After reading the manuscript a couple of times I miss a clear positioning about the hypotheses tested in the study. I can easily picture them, but they are not specifically presented in the last bit of the introductory section. For example, although it may seem obvious, do the authors hypothesize that similar microbiomes in the soil and caterpillars feeding on entire plants (not in detached leaves) imply that the latter community comes from the former?

We hypothesized that plants would acquire their microbiomes from the soil and that caterpillars would acquire their microbiomes from the plant. Hence, microbiomes of caterpillars would differ depending on the soil the plant has been growing in. We added the additional detached-leaf assay in sterile petri dishes as a 'control' for the study, cancelling out all environmental microbes, and thus ensuring that all caterpillar microbes were indeed derived from the leaf compartment.

We further hypothesized that there would be no differences between the microbiomes of caterpillars feeding detached leaves and caterpillars feeding on entire plants.

However, the design of the experiment also allowed us to test for similarities between

microbiomes of caterpillars and those of other compartments.

We have now added the hypothesis at the end of the introduction (L53-60).

2. Most of the conclusions of this study are based on comparisons among microbiome composition in the soil and in caterpillars reared either directly on the plant (probably in close contact with soil) or on detached leaves. Microbiomes of caterpillars feeding on entire plants are more similar to those of the soil than to microbiomes of caterpillars feeding on detached leaves. Microbiomes of caterpillars on entire plants are also more diverse than those of caterpillars on detached leaves. Based on these results the authors conclude that caterpillar microbiomes originate from the soil. I have two important issues with these results. The first one is that I am not sure the microbiome of caterpillars feeding on entire plants and on detached leaves are easily comparable. Larvae in entire plants were allowed to feed on plants during 14 days, but only 5 days on detached leaves. How can the authors rule out that observed differences are not just natural changes occurring along insect development? This has been found, for example, in *Spodoptera littoralis*: Chen, B. et al - 2016 - Biodiversity and activity of the gut microbiota across the life history of the insect herbivore *Spodoptera littoralis*. *Sci. Rep.*

Even though the experimental times were not the same, the larvae in both treatments were of similar size and developed until early L3 instars in all treatments when microbiomes were extracted.

We kept caterpillars on detached leaves for five days and on intact plants for 14 days because the caterpillars fed with detached leaves grew faster than the caterpillars on intact plants. After five days, we ran out of fresh fully expanded leaves that we could use for the detached leaf assay. Caterpillars on the intact plants were still much smaller (L2). Therefore, the intact-plant assay was continued until caterpillars reached the same larval instar. We speculate that the slower development of larvae on intact plants could be due to e.g. induced defense responses of intact plants. Larvae were visually compared and appeared very similar in size. Precise weighing, however, showed that larvae on the intact plants on average were 4 mg larger than larvae from

the detached leaf assay, although they were of the same larval instar (L3, Suppl. Fig 13). This difference is very small, since we know from other experiments with similar soil types and plants that larvae of this species grow at least to 200 mg or more before they pupate (R. Heinen, personal observations). Hence we believe it is unlikely that this can explain the differences we observed in microbiomes for the two groups of larvae. It should also be noted that none of the insects were deprived of food throughout the experiment. In the revised manuscript we now provide information about the size and comparison of the caterpillars in the two assays in the methods (L279-287) and also address the issue in the discussion (L185-194).

My second concern is that if caterpillars were allowed to freely move and feed on plants for 14 days, isn't it possible that the similarity between caterpillar and soil microbiomes are due to caterpillars "contaminating" the soil with their faeces? Moth's guts are quite alkaline and a hostile environment for microbes so it is possible that this explanation is more likely than what the authors propose. This could explain, for example, why some microbial taxa that are present in the soil are highly abundant in caterpillars. This issue would have been easily solved by feeding caterpillars on entire plants inside clip cages (not sure this is possible as I am not familiar with the biology of this moth). Alternatively, a soil control with plants, but without caterpillars (maybe imposing mechanical damage to plants to emulate herbivory) should be added.

The reviewer correctly points out that the caterpillars could have contaminated the soil through e.g. faeces and that their input may have influenced the composition of the soil microbiome. However, this does not explain the *differences* in microbiomes of **caterpillars on detached leaves** and **caterpillars on intact plants**, and the *similarities* we observed between **soil microbiomes** and **caterpillars on intact plants** only.

Moreover, we show in Suppl Fig. 6 and Fig. 2c that only a very small proportion of the microbiome is shared between both caterpillars kept on intact plants, caterpillars fed with detached leaves, and soils (this would be the fragment of 'unique caterpillar microbiome' that could end up in the soil).

This is a much smaller portion of the insect microbiome than the microbiome shared by only **caterpillars on intact plants** (with access to soil) and **the soil microbiome** itself. It is also important to note that caterpillar feces often remain on the leaves during experiments. The leaves that were in contact with caterpillars did not have significantly different microbiomes than the leaves from plants that were kept without caterpillars, which further suggests minimal effects of caterpillar feces on microbiomes.

Finally, as the insects weighed only 10 to 15 mg at the end of the experiment, and plants were grown in pots filled with 1 kg of soil, the amount of faeces (frass) was very small, and it is unlikely that the caterpillars produced enough faeces to have a considerable impact on soil microbiomes in this system. We have extended the discussion (L176-180) to take into consideration these suggestions.

3. Please see these references as insect herbivores have already been reported to acquire symbionts from the soil. Kikuchi Y, Hosokawa T, Fukatsu T (2011) Specific developmental window for establishment of an insect-microbe gut symbiosis. *Appl Environ Microbiol* 77:4075–4081. Kikuchi Y, Hayatsu M, Hosokawa T et al (2012) Symbiont-mediated insecticide resistance. *Proc Natl Acad Sci USA* 109:8618–8622

We thank the reviewer for pointing out these studies. The work of Kikuchi et al. shows that plant-feeding stinkbugs acquire symbiotic bacteria of the genus *Burkholderia*, from the soil. The novelty in our study is that we show that the entire bacterial and the fungal microbiomes of leaf feeding caterpillars resemble the microbiomes of the soil in which their host plant is growing, and, moreover, that soil legacies of previous plant growth are detected in the insect microbiome in the new plant-soil system. We have adjusted the introduction and discussion to incorporate the suggested relevant literature (L34-36 and L141-142).

Enric Frago - CIRAD

Reviewer #2 (Remarks to the Author):

This research focuses on how plant community alteration of the soil microbiome

influences the establishment of herbivore associated microbial communities. Soil from field plots was brought into the greenhouse to serve as a substrate to grow a common plant (dandelions). Herbivores (*Mamestra brassicae* caterpillars) were placed on these plants or on cut leaves from these plants. The researchers conducted an exhaustive set analyses of the relationships between the bacterial and fungal communities in the soil, on and in the plant leaves, and on and in the caterpillars. Their main finding is that caterpillars fed on intact plants are associated with a more diverse microbial community that is more similar to that of the soil than the caterpillars fed on cut leaves disassociated from the soil community. They interpret this finding as suggesting that caterpillar acquisition of microbes from the soil is an underrecognized component of how these herbivores establish their microbiomes. They further find that plant community composition alters the microbiome of soil, which in turns alters the microbiome of caterpillars reared in the presence of this soil.

There is a significant problem with the experimental design that makes me wary of the authors' main conclusion, namely that caterpillars fed on intact plants have a more diverse gut microbiome because they have access to the soil microbiome. That concern is that the caterpillars in the intact plant treatment were allowed to feed on plants for 14 days, ad libitum. The caterpillars in the cut leaves treatment were only fed on the plant leaves for 5 days, with the leaves needing to be changed every day. Therefore, the caterpillars in two treatments are different ages and likely different life stages. Furthermore, based on the descriptions, it is not clear if the cut leaf treatment caterpillars ran out of food some days, which would undoubtedly cause metabolic changes and stress signals that could alter the gut microbiome.

This remark was also made by the first reviewer and hence we refer to our answer above. In brief, the caterpillars were all from the same larval instar and similar in size. The insects were not deprived of food at any stage of the experiment. We have clarified this in the revised manuscript and in full detail above.

I am also concerned that larvae were fed on artificial diet until second instar, precluding establishment of the natural gut microbiome in the stage likely when most caterpillars would begin to establish this microbiome. This is particularly of concern if this diet contained antibiotics. The artificial diet recipe is not detailed in the methods, so I am not sure if antibiotics are a problem.

As correctly pointed out by the reviewer, the caterpillars were reared on artificial diet until the early second instar. It is also correctly pointed out that these diets contained a low amount of antibiotics (8g nipagin and 0.5g streptomycin added to 5L water and roughly 1500g of other ingredients; recipe provided in Suppl. Table S4). These antibiotics are primarily added to prevent the diet from being overgrown by molds in the first day.

The reviewer is right in saying that presence of these antibiotics may have suppressed the establishment of microbiomes. Nevertheless, our data clearly shows that microbiomes were abundantly present in caterpillars on intact plants and to lesser extent in those on detached leaves. Moreover, clear differences in caterpillar microbiomes were measurable, based on the history of plant growth on the donor soils in which *Taraxacum officinale* was grown.

We might even speculate that if we would have used freshly emerged caterpillars, in absence of the antibiotics, the patterns should even be stronger. Future studies should elucidate in which larval stage the uptake of microbiomes is highest. This has now been incorporated into the discussion.

I am also concerned that these results must be cautiously interpreted given that the main way that plant leaves acquire foliar endophytes is through rain shower, causing greenhouse grown plants to have few foliar fungal endophytes, for example, compared to outdoor reared plants. Thus, in nature, leaves likely have very different and more diverse microbial communities that likely shape the microbial composition of their insect folivores. Therefore, suggesting that soil is a more important driver of the insect microbiome in the plant leaves could be true in a greenhouse setting, but may not be true in nature.

We fully agree that under outdoor conditions microbiomes of leaves can be much more diverse than what was observed in our study (e.g. through aerosols, rain splash or interactions with passing animals). However, for our design it was essential to have a high level of control, which was necessary to provide a proof of concept. This is why we selected a greenhouse setting, even though we used field collected soils. We do not dispute that herbivorous insects can (and do) acquire at least part of their microbiomes from their host plants. In fact, that was what we hypothesized (this has now been made more explicit in our expanded intro).

We agree with the reviewer that we should not overemphasize these results and that it is important to further study these interactions in an ecologically relevant setting and now addressed this in the discussion (L172-176). However, we believe that our findings are important as they point out that we should not overlook that herbivorous insects, apart from ingestion through a plant-based diet, also acquire their microbiomes from the soil. We further emphasize that more studies are needed that examine how general these responses are and the importance of these soil-insect-microbiome interactions under field conditions (L182-184).

A final concern is that the caterpillar samples, as far as I can tell, are whole caterpillars rather than gut samples. While effort was made to surface sterilize these caterpillars, such efforts may not remove all residual DNA. Thus, the increased diversity of the intact plant associated caterpillars could be due to the fact that there were reared in a tent with a whole plant and soil rather than in a sterile Petri dish with a leaf. This could lead to more diverse communities on their exoskeleton, but these may not be functionally relevant. Most previous research on the microbiome of caterpillars has focused instead on the gut microbiome.

We thoroughly sterilized the caterpillars using hypochlorite and sonification, following protocols that are commonly applied for surface sterilization (Lundberg et al. 2012), for example of fine roots, which provide much more diverse and rich environments than caterpillars. Hence, it is highly unlikely that surface dwelling microbes were present in our microbiomes.

Moreover, in the soil we detected fungal species that are categorized as insect pathogens, such as *Beauveria* spp. and *Metarhizium* spp. (Sordariomycetes, Cordycipitaceae and Clavicipitaceae). These fungi are known to negatively affect the growth of insects and are usually associated with the insect cuticle¹). Interestingly, we found a negative correlation with the **relative abundance of these specific fungi in the soil microbiome and insect performance on intact plants**, suggesting that these fungi may have been present on the insect cuticle. However, we did not detect these fungi in the caterpillar microbiomes, which may indicate that they were removed successfully from the cuticle during the surface sterilization procedure.

1Shang, Y., Feng, P. & Wang, C. Fungi That Infect Insects: Altering Host Behavior and Beyond. *PLoS Path.* **11**, e1005037-e1005037, doi:10.1371/journal.ppat.1005037 (2015).

We now discuss this matter in the discussion (L194-201)

This manuscript should be carefully checked for grammar. For example, many sentences are missing commas, and phrases such as "low diverse" should be corrected.

We have carefully checked and proof read the entire manuscript.

Weight data should be made available in Dryad or another public database.

Data on biomass of caterpillars and plants are now included and all data will be made available open access upon acceptance.

Supplemental Figure 5. The taxa names in the legend should be simplified.

We have simplified the taxa names in the supplementary figure (now supplementary figure 6).

The abstract does not mention the insect species, a problem given that the title does not either.

We have not added the name of the insect species (*Mamestra brassicae*) and plant species (*Taraxacum officinale*) to the abstract since the word limit in the abstract had

to be considerably reduced to 150 words in the revision (originally the manuscript was transferred from submission to Nature). However, if needed we are willing of course to add the Latin names of the species to the abstract.

Reviewer #3 (Remarks to the Author):

The manuscript by Emilia Hannula et al. seeks to answer which are the forces that drive microbiome composition in plant-feeding insects. They found that caterpillars get most of their microbiome from the soil as opposed to the plants they eat, and that previous history of plants grown in those soils also influence caterpillar microbiome composition. I enjoyed reading the manuscript and I think it presents a series of relevant results from well-designed experiments which will be appreciated by a broad scientific audience. However I have some concerns which I list below:

Major comments:

1. My major concern is the lack of any form of multiple testing correction. The authors provide many examples of differential diversities, taxa and correlations with nominally significant p-values, but none of them seem to have been adjusted for multiple testing. This adjustment must be done throughout the manuscript to allow for interpretation of the results.

We did perform Benjamini, Hochberg, and Yekutieli (Benjamini & Hochberg 1995²) corrections on all p-values (FDR-correction), but erroneously did not mention this in the paragraph with statistical tests. This has now been corrected in the revised version and we mention that the p-values are FDR corrected.

²Benjamini, Y., and Hochberg, Y. (1995). Controlling the false discovery rate: a practical and powerful approach to multiple testing. *Journal of the Royal Statistical Society Series B*, 57, 289–300.

This has now been clarified in the (extended) methods (L370-374)

2. My search for accession number 'PRJEB27512' at the EBI-ENA got no results.

Please ensure that the data is made publicly available. Also, it is important that all the relevant sample metadata is made available.

The data has already been submitted to the ENA site. We have now included in the submission files for the manuscript, the metadata file, and the transcript of the submission to ENA. It is common policy that the data will be made publicly available after acceptance of the manuscript and we fully comply with this. However, at this stage we are, of course, prepared to provide the raw data (which consists of a number very large files) to the reviewers if they would request this. We have enquired at ENA whether it is possible to allow reviewers to access the files that are not yet made publicly available, but unfortunately (and surprisingly) this is not yet possible.

3. The authors present many compelling visualizations and summary statistics showing that the caterpillar microbiota comes from the soil. While they are all consistent with the author's interpretation, they are also indirect. Bray-Curtis focuses on relative abundances differences between shared taxa, and therefore one can easily construct toy examples of NMDS plots like the ones of fig 1 where the OTU sharing is identical between the groups. The ternary plots from fig 2 are also not ideal, since they show relative abundance ratios and not membership. I think a more direct test of the author's claim would be an analysis of OTU prevalence. For example, how many OTUs are present in almost all intact plant caterpillar and soil samples but never in leaves and detached leaves caterpillars? And what proportion of the community these OTUs make? A clustering and heatmap with presence absence data might also be useful

We agree that differences between categories in NMDS with Bray-Curtis distance can be due to either a) changes in relative abundances of OTUs that are shared or b) differences in OTU composition. We detect both. As seen from figures 1A and 1B (and supplementary figure 2), at the class and phylum level there is a large difference between the number of OTUs present in the samples and at the same time in relative abundances of the classes and phyla. The microbiome in the caterpillars reared on detached leaves had relatively more Gamma- and Betaproteobacteria. In Fig 2 (and

especially Fig 2C) we show which OTUs are shared between compartments. This figure also answers the question of the reviewer regarding the number of OTUs that are present in caterpillars on intact plants and soil samples but never found in leaves and caterpillars on detached leaves. The identity of these OTUs is further shown in Supplementary Fig 6. We now added to text the percentage of unique OTUs in caterpillars (78% unique for caterpillars on intact plants, 21% in both types of caterpillars (caterpillar core microbiome) and <1% unique for caterpillars fed on detached leaves). We also now included a heatmap with presence/absence data (OTU present in % of samples) as suggested (Suppl. Fig. 7). In the heat map it can be seen that also with this distance measure the caterpillars on intact plants cluster closest to soil while caterpillars on detached leaves closest to soil. Showing heat-maps of each individual sample is not feasible due to the large number of samples.

4. The authors tested alpha-diversity using linear models, however the model specification is not provided. Was it a Gaussian model? Which variables were included? Was any of the variables transformed? More importantly, it is not clear if the authors controlled for potential systematic differences in sampling depth. Cumulative sum scaling was benchmarked only for metrics of beta-diversity, therefore it is important to include terms for sequencing depth in tests for differences of alpha-diversity.

We indeed used a Gaussian model (GLM). We have now added details on the construction of the models in the statistical analysis paragraph in the methods. To fulfill assumptions of normality in the data, Chao1 data was ln transformed. Information on transformation is added to the Materials and Methods section. We did test for the differences in sampling depth across treatments and categories. Sampling depth was fairly even across the samples in each category and the extremes were filtered out (3*standard deviation filtering) per sample category (this is reported in Supplementary Table 5). The sampling depth (read numbers) was added to the model as a covariant but did not change the outcome.

Detailed descriptions of the statistical models used can now be found in the methods (L359-370).

5. The authors tested the abundance of many specific microbial taxa across different treatment groups. The model used was not specified here either so the same clarifications as above are needed here. Moreover, the standard approach would be to use a form of negative binomial model that accounts for sequencing depth differences, overdispersion and that models mean-variance relationships accurately. It appears that the authors performed some type of gaussian ANOVA, which is not really appropriate for count data.

We have now provided more information about the statistical tests that we used. We agree with the reviewer that the data has a zero-inflated beta distribution. To account for the overdispersion in the model, we used Poisson distributions in our generalized linear model (GLM) for all count data. We furthermore fitted the zero-inflated Poisson regression models (package PSCL in R) but with our data they were not superior to GLM with Poisson distributions (Vuong test $p > 0.05$). The full model we used is as specified as follows in R:

```
Model2 <- glm(x ~ y1 + y2 + (1|Plot) + offset(read.number), family = poisson, data = ourdata))
```

```
(the zero inflated model was: model1 <- zeroinfl(x ~ y1 + y2 |NumberOfReads , data = ourdata)).
```

where x = individual bacterial phyla or fungal class and y_1, y_2, \dots = compartment (e.g. leaves, soil, roots, caterpillars on intact plants or detached leaves) and/or legacy of soil.

(Vuong Non-Nested Hypothesis Test-Statistic:

(test-statistic is asymptotically distributed $N(0,1)$ under the null that the models are indistinguishable)

Vuong z-statistic

H_A p-value

Raw	-1.540593 model2 > model1 0.061708
AIC-corrected	-1.344066 model2 > model1 0.089463
BIC-corrected	-1.058584 model2 > model1 0.144895)

From this model we estimated significance based on a Chi-square test and corrected the p-values for multiple comparisons. The new analyses are now presented and discussed throughout the revised version of the manuscript. Detailed descriptions of the statistical models used can now be found in the methods (L359-370).

6. In both, test of diversity and microbial abundances, only some comparisons are highlighted with a p-value in the visualizations, but it is not clear which tests were actually performed. Did the authors performed all pairwise comparisons? ANOVA followed by Tukey? The authors should clarify which tests were performed and correct for multiple testing appropriately.

We have now added details about the model used and specified the corrections used for false discovery rates (FDR). We now also report in the figures if the full model is significant, i.e. if the factor significantly affects the variable (bacterial phylum or fungal class) based on a Chi-square test from the GLM. We base our pair-wise comparisons on a Tukey test following the GLM (Tukey = emmeans(Model2, ~ TypePlace), pairs(Tukey)).

7. I didn't understand if the caterpillars are genetically identical. I guess not. Given the high replication, I don't think this challenges the results but the authors should be explicit about this.

The caterpillars were mass reared and hence were not all genetically identical. All eggs originated from one batch (usually 1000-3000 eggs), but the eggs were laid by many different moth females in the rearing. The caterpillars that we used in the experiment all were randomly selected from this single batch. We have now clarified this in the manuscript in the methods (L255-259).

8. One of the main conclusions is that caterpillars get their microbiota from the soil, but little discussion is given as to how this might happen. Is this simply because they occasionally walk on the soil? Or is the transfer mediated by the plant or air? Moreover, I wonder how the authors reconcile this observation with previous work that claims that caterpillar microbiome is mostly transient. how biologically relevant is the caterpillar microbiota for the caterpillars themselves?

We have expanded the discussion on this topic. Caterpillars indeed walk on the soil and we observed this behavior also in previous studies. We consider this the most likely mechanism of uptake from the soil. We now also refer to the work of Kikuchi et al. about uptake of *Burkholderia* by stinkbugs from the soil and place our work into the context of transient microbiomes of insects and link to the work of Hammer et al. 2017 in PNAS (see L154-161).

9. The authors find that some microbes are associated with caterpillar performance mostly when fed on intact plants growing in conditioned soil. I wonder how likely are the performance benefits due to the microbes themselves versus some abiotic edaphic factor like micronutrient composition, which could be mediated by the plant. Did the authors perform micronutrient analysis in the conditioned soils? Or in the plant tissue? Was there any plant performance difference? And did plant performance correlate with caterpillar performance?

The reviewer brings to attention an important point that we could actually test, namely that the observed soil effects on caterpillar performance would be caused by abiotic factors mediated by the plants. We now include data that show that soils affect plant growth in a very similar way in both parallel assays (L113-116 and Suppl. Fig. 9). Moreover, we now include data that show that soil origin had a clear effect on caterpillar biomass (L 116-118 and Suppl. Fig. 10). However, this effect was only observed when caterpillars were feeding on intact plants, whereas soil origin had no effect in the detached leaf assay. As plants were equally affected in both assays, one would expect that plant chemistry would thus also be affected in a similar way.

Further, we performed nutrient analyses on the donor field soils, which are now included in the manuscript (L 118-122). These analyses show that most nutrients in the soil are not affected by the plant communities that grow on the soils. Only nitrogen availability was higher in soils conditioned by grass communities than in other soils (Suppl. Table S1, Suppl. Fig. 11). However, we observed no relationships between individual soil nutrients and caterpillar or plant performance parameters (Suppl. Fig. 12). All arguments taken into account, we now provide strong evidence that soil microbiomes do influence caterpillar microbiomes and that this, in turn, leads to differences in performance of the caterpillar.

Other comments

1. The resolution of the figures in my review copy was low. Please make sure you have high resolution versions for the actual publication.

We made sure the resolution is high enough and we made vector images which we now supply in .tiff format.

2. Colors for soil and root were hard to distinguish in my printed copy. Make sure you pick colors that are easily differentiable.

We have now adjusted colors so that soil and root are more distinguishable.

3. Axis labels are really small in many figures, for example in taxonomic abundance plots.

We enlarged the axes in all figures.

4. For figure 4, I recommend indicating more clearly in the figure which rows/columns are caterpillar metrics, and which are microbial abundances

We clarified this by coloring the text brown for caterpillar metrics

5. I like supp figure 8, but it would be better if the diagram also included the creation of different soil legacies with different combinations of plants. It would also be nice if

the diagram included the number of replicates of each class.

We now expanded this figure (which is Suppl. Fig. 1 in the revised version) and include information about the soil legacies that were used as donor soils.

6. Line 159 of methods: there is a typo 'co-occurance' should be 'co-occurrence'

Corrected.

REVIEWERS' COMMENTS:

Reviewer #1 (Remarks to the Author):

This is the second time I read this manuscript, which is as I mentioned in my previous revision, exciting, well written and presents an impressive amount of data. Most of my main concerns have been addressed by the authors. I had some concerns on the methods used, and the authors justify them based on the biology of the studied species and differences in microbe diversity and composition found. These explanations are somehow satisfactory, please see my comments below.

One of my main concerns was that larvae were allowed to feed during 14 days on entire plants, but only 5 days on detached leaves, which makes microbiome comparisons difficult. As explained in their rebuttal letter and in the manuscript, it seems that caterpillars on detached leaves develop much faster so that by the end both groups reach similar sizes (and instars). The authors therefore assume that larvae in both groups are in the same developmental stage allowing for microbiome comparisons, which is a fair argument. The non-comparability of both groups due to differences in developmental time is still an issue, but probably minor (I am not an expert on the biology of the insect used).

My second concern was that if caterpillars were allowed to freely move and feed on plants for 14 days, it is possible that the similarity between caterpillar and soil microbiomes were due to caterpillars "contaminating" the soil with their faeces. Based on this problem, I proposed to add a soil control with plants, but without caterpillars. This issue, together with the issue that caterpillars can also acquire microbes from the soil and not directly from the plant, has also been raised by the other two reviewers. These shared concerns indicate a potential major weakness of the present study. In response to these questions the authors argue that caterpillar-to-soil or soil-to-caterpillar transmission is unlikely because of the differences found between microbiomes of caterpillars on detached leaves and caterpillars on intact plants, and the similarities observed between soil microbiomes and caterpillars on intact plants. As with my previous concern, this is a fair point. I agree, however, with one of the other reviewers that comparing community composition remains an indirect way to test symbiont acquisition through the plant. Maybe the authors should acknowledge this in their discussion section, and propose future experiments that could provide a stronger evidence to the patterns found here. For example, would it be possible to mark with isotopes soil microbes and follow their way into insects?

Enric Frago, CIRAD

Reviewer #2 (Remarks to the Author):

I have reviewed the revised manuscript and read the comments to the reviewer's suggestions carefully. I still believe that there are several important limitations of the study design that make it hard to draw conclusions from the data. These are:

- 1) I still believe that the comparison between the microbiomes of caterpillars fed on intact plants and detached leaves provides non-informative data. These differences could be due to developmental differences, which appear to be extreme in terms of developmental rate, due to changes in plant chemistry that then alter the insect microbiome, or due to the ability of the insects to more easily directly contact the soil microbes, either because they interact with the soil while wandering, or because during the watering process the soil microbes could get onto plant surfaces. In other words, given the many ways in which the intact plant and detached leaf treatments differ, I do not think you can "disentangle the effects of the soil microbiome on the caterpillar microbiome mediated via the

plant from the possible direct effects via the soil" (line 62).

2) In a natural setting, dynamics could be different. It is well known that plants acquire symbionts from a variety of contexts, for example rain water. Under non-greenhouse conditions. The more natural communities associated with leaf tissue could play a bigger role in shaping herbivore microbiomes. You now mention this in the discussion, but you still make broad conclusions in your abstract and discuss that do not reflect this limitation of your experimental design.

3) Given that these caterpillars were exposed to antibiotics prior to the experimental treatments, these results would be more informative if you knew if the bacterial titer, particularly in the gut, was at a level similar to what is found in caterpillars not fed antibiotics. If the titer was substantially lower, then it would suggest that differences could be driven by small changes in abundance of taxa, which could be due to surface contaminants that were not removed with sterilization. While you state in your response and in the discussion that you used accepted methods to remove all surface contaminants, the paper you cite to prove the efficacy of this sterilization process is a paper not on insects but on plants. Thus, you cannot state with confidence that surface contaminants are not an issue.

Minor point. The added paragraphs in the discussion in response to reviewer concerns may be hard to follow for the reader as the points are not well connected with one another.

Reviewer #3 (Remarks to the Author):

I read with interest the improved manuscript by Emilia Hannula et al. I reiterate my opinion that this work presents original and relevant results that will be of interest to a variety of scientists.

I appreciate the clarifications regarding the statistical methods and data availability. I find the methods appropriate, and the description sufficient for other scientists to evaluate and reproduce. I also find the new color palette much easier to follow.

I have a couple of general comments left.

It is not clear to me that plant-soil feedbacks are being measured. The experiment is designed to test the hypothesis (now stated in the introduction) that there is a below-ground to above ground effect on caterpillar microbiome mediated by the plant. The conclusion is that there is an effect, but it is not mediated by the plant. It was already known that growing different plants can change the soil microbiome. Therefore, the novelty is on the direct effect of soil on the caterpillar microbiome. Given the described experiments, I would expect similar results if the authors had used different soils (as opposed to the same soil but differentially conditioned), or if they had used some abiotic treatment for conditioning the soil. Insects can influence plant, and there is a compelling hypothesis to be made regarding plant-soil feedbacks, but the current experiments cannot confirm nor discard that hypothesis. I think a more appropriate title would be something like "Soil directly alters microbiomes of foliar-feeding insects", and the authors should be more careful when talking about plant-soil feedback on the discussion.

The other reviewers raised the point that the microbiomes from the two different assays are not harvested at the same time point, which could explain the differences. The authors argue that developmental differences are unlikely to be responsible for the observed differences. I am not familiar with this system but I find the author's arguments reasonable. However, an alternative -

overlooked- explanation for the observed differences could be temporal dynamics. If the microbial communities in the different assays are not in steady state, and if the major force driving microbiome composition is microbe-microbe competition, then one can obtain different communities based on sampling time regardless of the effect (or absence of effect) of developmental stage. Is there any evidence, from the authors or previous work, that the caterpillar communities are at steady state at the different sampling times?

REVIEWERS' COMMENTS:

Reviewer #1 (Remarks to the Author):

This is the second time I read this manuscript, which is as I mentioned in my previous revision, exciting, well written and presents an impressive amount of data. Most of my main concerns have been addressed by the authors. I had some concerns on the methods used, and the authors justify them based on the biology of the studied species and differences in microbe diversity and composition found. These explanations are somehow satisfactory, please see my comments below.

One of my main concerns was that larvae were allowed to feed during 14 days on entire plants, but only 5 days on detached leaves, which makes microbiome comparisons difficult. As explained in their rebuttal letter and in the manuscript, it seems that caterpillars on detached leaves develop much faster so that by the end both groups reach similar sizes (and instars). The authors therefore assume that larvae in both groups are in the same developmental stage allowing for microbiome comparisons, which is a fair argument. The non-comparability of both groups due to differences in developmental time is still an issue, but probably minor (I am not an expert on the biology of the insect used).

Agreed

My second concern was that if caterpillars were allowed to freely move and feed on plants for 14 days, it is possible that the similarity between caterpillar and soil microbiomes were due to caterpillars "contaminating" the soil with their faeces. Based on this problem, I proposed to add a soil control with plants, but without caterpillars. This issue, together with the issue that caterpillars can also acquire microbes from the soil and not directly from the plant, has also been raised by the other two reviewers. These shared concerns indicate a potential major weakness of the present study. In response to these questions the authors argue that caterpillar-to-soil or soil-to-caterpillar transmission is unlikely because of the differences found between microbiomes of caterpillars on detached leaves and caterpillars on intact plants, and the similarities observed between soil microbiomes and caterpillars on intact plants. As with my previous concern, this is a fair point.

We already addressed this issue in the discussion but have highlighted this caveat more clearly now. We also explained in the previous rebuttal that the caterpillars that we analyzed were very small and hence that any potential impact of the caterpillars on the soil would be marginal. This is also mentioned in the discussion.

I agree, however, with one of the other reviewers that comparing community composition remains an indirect way to test symbiont acquisition through the plant. Maybe the authors should acknowledge this in their discussion section, and propose future experiments that could provide a stronger evidence to the patterns found here. For example, would it be possible to mark with isotopes soil microbes and follow their way into insects?

Following the suggestion of the reviewer, we now mention in the discussion that studies with labeled soil microbes should further examine the direct and indirect interactions between soil, plant and insect microbiomes.

Enric Frago, CIRAD

Reviewer #2 (Remarks to the Author):

I have reviewed the revised manuscript and read the comments to the reviewer's suggestions carefully. I still believe that there are several important limitations of the study design that make it hard to draw conclusions from the data. These are:

1) I still believe that the comparison between the microbiomes of caterpillars fed on intact plants and detached leaves provides non-informative data. These differences could be due to developmental differences, which appear to be extreme in terms of developmental rate, due to changes in plant chemistry that then alter the insect microbiome, or due to the ability of the insects to more easily directly contact the soil microbes, either because they interact with the soil while wandering, or because during the watering process the soil microbes could get onto plant surfaces. In other words, given the many ways in which the intact plant and detached leaf treatments differ, I do not think you can "disentangle the effects of the soil microbiome on the caterpillar microbiome mediated via the plant from the possible direct effects via the soil" (line 62).

The reviewer correctly points out that the two parallel assays differed in duration. This was not as planned initially, as they should have had the same duration. However, as we explained in the previous rebuttal, this was a choice that we had to make, as the caterpillars belonged to a different instar on the intact plants (and this is known to influence their microbiomes), when the detached leaf assay was terminated (due to shortage of suitable leaves). The caterpillars were thus kept on the plant until they reached a (visually) similar size, and same instar. We disagree with the reviewer that differences in microbiomes of the insects are directly due to induced plant defense as the microbiomes of the insects on intact plants did not resemble the plant microbiome but the soil microbiome. Induced plant defense may have caused the insects to wander off the plant more often and hence result in insects having more contact with the soil, but this is not in contrast with what we propose.

Watering was done carefully and at soil level, and did not result in splashing from the soil onto the leaves. Moreover, in contrast to what the reviewer suggests, if the microbiomes of the insects would have been altered via eating leaves (and not via contact with the soil) on which soil microbes were present due to splashing after watering, we would expect that this would influence both groups of caterpillars since these were both eating leaves and all plants were watered in the same way.

We conclude from our study that.

- 1) plants have a fairly limited microbiome under controlled conditions relative to the soil microbiome.
- 2) on intact plants, the caterpillar microbiome resembles the microbiome from the leaf to a certain extent but there is a much higher resemblance with the soil microbiome.
- 3) this is not the case for insects on detached leaves, where there is only resemblance with the leaf microbiome. Thus, we show that insects when they have access to soils, have much richer microbiomes, and that these resemble the microbiome of soils and the legacies of previously growing plants.
- 4) several groups of soil microbes show relationships with the biomass of the insects also suggesting a direct link between soil and insect microbiomes.

2) In a natural setting, dynamics could be different. It is well known that plants acquire symbionts from a variety of contexts, for example rain water. Under non-greenhouse conditions. The more natural communities associated with leaf tissue could play a bigger role in shaping herbivore microbiomes. You now mention this in the discussion, but you still make broad conclusions in your abstract and discuss that do not reflect this limitation of your experimental design.

We fully agree with the reviewer that in natural settings dynamics could be different, and have now added this to the discussion. We do not agree with the reviewer however that the conclusions in the abstract are too broad. We only state in the abstract that insect microbiomes depend on soil microbiomes and that effects of plants on soil microbiomes can be transmitted to aboveground insects feeding later on other plants. This is exactly what our study shows.

3) Given that these caterpillars were exposed to antibiotics prior to the experimental treatments, these results would be more informative if you knew if the bacterial titer, particularly in the gut, was at a level similar to what is found in caterpillars not fed antibiotics. If the titer was substantially lower, then it would suggest that differences could be driven by small changes in abundance of taxa, which could be due to surface contaminants that were not removed with sterilization. While you state in your response and in the discussion that you used accepted methods to remove all surface contaminants, the paper you cite to prove the efficacy of this sterilization process is a paper not on insects but on plants. Thus, you cannot state with confidence that surface contaminants are not an issue.

We have altered these statements made in the discussion to further emphasize these concerns.

Minor point. The added paragraphs in the discussion in response to reviewer concerns may be hard to follow for the reader as the points are not well connected with one another.

We have changed the wording in these paragraphs.

Reviewer #3 (Remarks to the Author):

I read with interest the improved manuscript by Emilia Hannula et al. I reiterate my opinion that this work presents original and relevant results that will be of interest to a variety of scientists.

I appreciate the clarifications regarding the statistical methods and data availability. I find the methods appropriate, and the description sufficient for other scientists to evaluate and reproduce. I also find the new color palette much easier to follow.

I have a couple of general comments left.

It is not clear to me that plant-soil feedbacks are being measured. The experiment is designed to test the hypothesis (now stated in the introduction) that there is a below-ground to above ground effect on caterpillar microbiome mediated by the plant. The conclusion is that there is an effect, but it is not mediated by the plant. It was already known that growing different plants can change the soil microbiome. Therefore, the novelty is on the direct effect of soil on the caterpillar microbiome. Given the described experiments, I would expect similar results if the authors had used different soils (as

opposed to the same soil but differentially conditioned), or if they had used some abiotic treatment for conditioning the soil. Insects can influence plant, and there is a compelling hypothesis to be made regarding plant-soil feedbacks, but the current experiments cannot confirm nor discard that hypothesis. I think a more appropriate title would be something like “Soil directly alters microbiomes of foliar-feeding insects”, and the authors should be more careful when talking about plant-soil feedback on the discussion.

A similar point was raised by the journal editor and we have decided to change the title, in accordance with her suggestion, to “Foliar-feeding insects acquire microbiomes from the soil rather than the host plant”.

We agree that it is known that plants changing soil microbiomes. However, the novelty in this study is that these plant-induced changes in soil microbiomes are taken up and reflected in the caterpillar microbiome that feeds on a plant that grows later in this soil. We have changed terminology from plant-soil feedback to (microbial) soil legacy effects, to avoid potential confusion.

The other reviewers raised the point that the microbiomes from the two different assays are not harvested at the same time point, which could explain the differences. The authors argue that developmental differences are unlikely to be responsible for the observed differences. I am not familiar with this system but I find the author’s arguments reasonable. However, an alternative -overlooked- explanation for the observed differences could be temporal dynamics. If the microbial communities in the different assays are not in steady state, and if the major force driving microbiome composition is microbe-microbe competition, then one can obtain different communities based on sampling time regardless of the effect (or absence of effect) of developmental stage. Is there any evidence, from the authors or previous work, that the caterpillar communities are at steady state at the different sampling times?

Indeed, temporal aspects may play a role in the composition of a (or any) microbial community. We would assume that microbial communities are rarely steady. Our work thus far (mostly on soil and rhizosphere microbiomes) would confirm this, for both fungi and bacteria. Microbiomes in insects have also been shown to differ over time (e.g. larval stages). However, the microbiomes of caterpillars and soils of the intact plant assays were both sampled on the same sampling day. We would thus assume that these time effects were minimal.

It is important to notice that the microbiome of the leaves in both assays did not show any resemblance to the soil microbiomes, but that the insect microbiomes did resemble the soil microbiome and that legacies of earlier plant growth in the soil microbiome were detected in the insect microbiome. Since these legacies were not detected in the leaves, it is not likely that these legacies would appear in the caterpillars that fed detached leaves, if the caterpillars would have been kept for several days longer in the petri dishes. This suggests that the role of the plant microbiome is limited for insect microbiomes, especially when they are in contact with other rich environmental compartments, such as soil.

Sur Herrera Paredes